# ECHO: Entropy-Confidence Hybrid Optimization for Test-Time Reinforcement Learning

Chu Zhao [* 1]   Enneng Yang [* 2]   Yuting Liu [1]   Jianzhe Zhao [1]   Guibing Guo [1]

## Abstract

Test-time reinforcement learning generates multiple candidate answers via repeated rollouts and performs online updates using pseudo-labels constructed by majority voting. To reduce overhead and improve exploration, prior work introduces tree-structured rollouts, which share reasoning prefixes and branch at key nodes to improve sampling efficiency. However, this paradigm still faces two challenges: (1) high-entropy branching can trigger rollout collapse, where the branching budget concentrates on a few trajectories with consecutive high-entropy segments, rapidly reducing the number of effective branches; (2) early pseudo-labels are noisy and biased, which can induce self-reinforcing overfitting, causing the policy to sharpen prematurely and suppress exploration. To address these issues, we propose Entropy–Confidence Hybrid Group Relative Policy Optimization (ECHO). During rollout, ECHO jointly leverages local entropy and group-level confidence to adaptively control branch width, and further introduces online confidence-based pruning to terminate persistently low-confidence branches, avoiding high-entropy traps and mitigating collapse. During policy updates, ECHO employs confidence-adaptive clipping and an entropy–confidence hybrid advantage shaping approach to enhance training robustness and mitigate early-stage bias. Experiments demonstrate that ECHO achieves consistent gains on multiple mathematical and visual reasoning benchmarks, and generalizes more effectively under a limited rollout budget. The implementation code is available at https://github.com/user683/ECHO

*Equal contribution   [1]Northeastern University, Shenyang, China   [2]Shenzhen Campus of Sun Yat-sen University, China. Correspondence to: Guibing Guo and Jianzhe Zhao <guogb,zhaojz@swc.neu.edu.cn>.

*Proceedings of the 43rd International Conference on Machine Learning*, Seoul, South Korea. PMLR 306, 2026. Copyright 2026 by the author(s).

## 1. Introduction

Current language models (Liu et al., 2024; Bai et al., 2023; Yang et al., 2025) often rely on high-quality human-labeled data for preference alignment, yet in many scenarios such labels are difficult to obtain or prohibitively expensive. To address this, researchers have proposed Test-Time Reinforcement Learning (TTRL) (Zuo et al., 2025), which improves reasoning capability by optimizing on self-feedback signals generated during inference, without requiring external supervision. In a typical TTRL setup, the model produces multiple candidate solutions for the same query, constructs pseudo-labels via self-consistency (Zhao et al., 2025a) or majority voting, and performs online updates accordingly. However, this paradigm is highly sensitive to the rollout budget: to obtain sufficiently stable pseudo-labels, conventional chain-based rollouts often require dozens to hundreds of samples, incurring substantial parallel sampling costs. To reduce this overhead and improve exploration, recent works (Liu et al., 2025) introduce tree-search-based rollouts (Ji et al., 2025; Hou et al., 2025), which share reasoning prefixes and branch at key decision points to achieve broader and more efficient sampling coverage under the same token budget, thereby producing more reliable learning signals at lower cost.

Despite the effectiveness of the above approaches, existing TTRL methods still face two key challenges in practice. **First, high-entropy branching can trigger rollout collapse**. ETMR (Liu et al., 2025) branches at high-entropy nodes, but under majority-vote pseudo-labeling, the branching budget is repeatedly consumed by a small number of trajectories with consecutive high-entropy steps. As a result, the number of effective branches quickly shrinks, the search tree degenerates into an almost chain-like rollout, and pseudo-labels become dominated by homogenized trajectories ultimately reducing exploration coverage and degrading the quality of learning signals. **Second, early pseudo-label bias leads to self-reinforcing overfitting**. In the early stage of training, pseudo-rewards are noisy and biased, which can push the policy to prematurely converge to a locally "high-scoring" solution and form a feedback loop: the output distribution rapidly sharpens (entropy drops), exploration collapses, and both later-stage performance and generaliza-

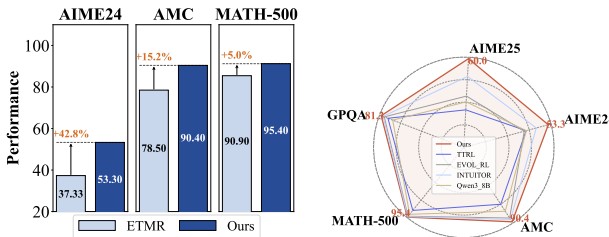

*Figure 1.* Performance overview of ECHO algorithm.

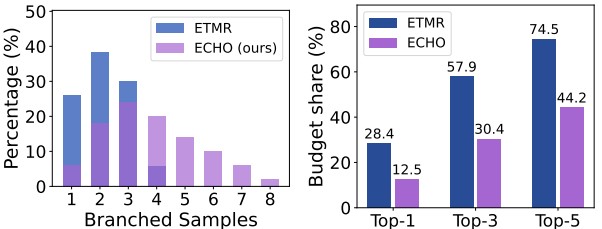

*Figure 2.* Empirical study of high-entropy rollout collapse in entropy-driven tree search for TTRL.

tion are constrained. Following the existing work (Dong et al., 2025), we conduct two types of statistics over ETMR rollouts: (i) high-entropy continuity, which measures the proportion of consecutive high-entropy segments and their length distribution within each trajectory, to characterize whether high-entropy steps persistently cluster along local paths; and (ii) branch budget allocation, which summarizes how the fixed branching budget is distributed across trajectories (e.g., the number of effective branches per batch and the Top-3 budget share), to quantify budget concentration and identify rollout collapse. The results in Figure 2 further validate this phenomenon: ETMR's branching budget is heavily skewed toward a few trajectories, so most rollouts effectively cover only 1–3 branches, exhibiting clear branching imbalance. The right panel shows that the majority of the budget is dominated by the top-ranked trajectories, while the remaining candidates receive almost no exploration resources, thereby weakening sampling coverage and reducing pseudo-label diversity.

To address the above challenges, we propose Entropy and Confidence Hybrid Group Relative Policy Optimization (ECHO). For Challenge 1, we introduce an entropy-confidence hybrid tree search. During node expansion, ECHO jointly models local entropy and group-level confidence, and dynamically balances them with adaptive weights. It shrinks the branching width in high-confidence regions to suppress unproductive expansions, while increasing the branching width in high-entropy and low-confidence regions to strengthen exploration and improve diversity. To prevent the search from being persistently trapped in highly uncertain high-entropy traps, we further incorporate online confidence-based pruning. Specifically, we use the minimum confidence within a sliding window along each trajectory as a quality criterion, and immediately terminate and prune a branch when it falls below an adaptive threshold estimated by warm-up quantiles. This mechanism distinguishes reasonable transient exploration from persistently low-confidence erroneous reasoning chains, thereby mitigating high-entropy collapse and improving both reasoning efficiency and trajectory quality. For Challenge 2, ECHO applies confidence-adaptive clipping to couple the update magnitude with sample reliability, preventing noisy pseudo-labels from inducing overly aggressive updates. In addition, we introduce entropy and confidence joint advantage shap-

ing, which increases the learning weight on tokens from effective but low-confidence trajectories. This design enhances training robustness and alleviates early-stage bias-induced overfitting and entropy collapse. As shown in Figure 1, ECHO consistently outperforms strong baselines on representative reasoning benchmarks, demonstrating robust gains on both mathematical and general QA tasks. The radar plot further confirms that these improvements are broad and balanced across all evaluated datasets, rather than concentrated on a single benchmark.

Our contributions are summarized as follows: **(1)** We propose Entropy–Confidence Hybrid Group Relative Policy Optimization, which jointly regulates local entropy and group-level confidence during rollout via adaptive weighting, and incorporates online pruning based on sliding-window confidence and warm-up quantile thresholds to suppress persistently low-confidence branches and alleviate rollout collapse. **(2)** We introduce the ECHO update mechanism, which combines confidence-adaptive clipping with entropy–confidence joint advantage shaping to reduce the impact of noisy pseudo-labels and to strengthen learning from tokens on effective yet low-confidence trajectories, thereby mitigating early-stage bias and entropy collapse while improving robustness and generalization. **(3)** Extensive experiments on mathematical and visual reasoning benchmarks demonstrate the method's consistent gains.

## 2. Preliminaries

Recent studies (Zhao et al., 2025b; Liu et al., 2025; Fu et al., 2025) have moved away from external annotations and instead exploit model-intrinsic statistics from token probability distributions to score, filter, and guide reasoning trajectories. Two widely used signals are: **(1) Token Entropy.** Given the predicted distribution $P_i$ at position $i$, token entropy is

$$H_i = -\sum_j P_i(j) \log P_i(j), \quad (1)$$

where $P_i(j)$ denotes the probability of the $j$-th vocabulary token. Lower $H_i$ indicates a sharper (more certain) distribution. **(2) Token Confidence Score.** Let $\text{Top}k(P_i)$ denote the set of top-$k$ tokens ranked by $P_i(\cdot)$. We define a token-

level confidence score as the mean probability mass over these candidates:

$$C_i = \frac{1}{k} \sum_{v \in \text{Top}k(P_i)} P_i(v), \qquad (2)$$

where $k$ is a fixed hyperparameter. Larger $C_i$ indicates higher token-level confidence. Building on these signals, prior work (Liu et al., 2025) adopts tree-structured rollouts in test-time RL, typically branching at high-entropy positions to improve diversity under a limited budget. However, high entropy-only branching often suffers from **high-entropy collapse**, where rollout budget over-concentrates on persistently uncertain regions, resulting in imbalanced exploration and degraded trajectory quality. We address this by an **entropy–confidence jointly optimized** tree-search framework that co-regulates uncertainty and confidence for branch triggering and budget allocation.

## 3. Methodology

This section details ECHO. Section 3.1 presents an entropy–confidence guided tree-structured rollout, Section 3.2 introduces confidence-adaptive clipping for stable updates under noisy pseudo-rewards, and Section 3.3 proposes entropy–confidence hybrid advantage shaping to emphasize uncertain yet decision-critical tokens. Figure 3 overviews the framework.

### 3.1. Tree-Structured Rollout with Entropy–Confidence Hybrid Optimization

In this section, we introduce an entropy–confidence guided tree-structured rollout. It uses token-level entropy and confidence, with window-based smoothing, to adaptively decide where to branch and how many branches to allocate. Online pruning further terminates degenerate branches early, alleviating high-entropy collapse and improving rollout efficiency under a limited inference budget.

**Entropy Increment and Window Smoothing.** Existing work (Fu et al., 2025) leverages token-level confidence to prune chain-based rollouts: when confidence stays low or shows a clear degradation trend, generation is terminated early to avoid wasting tokens and to improve estimation efficiency. Inspired by this work, we jointly model token entropy and token-level *confidence*, and use window-based smoothing to obtain stable signals for deciding when to branch and how wide to branch in tree-structured rollouts. Concretely, to avoid single-step noise, we apply temporal smoothing to both confidence and entropy. Let $W_G$, $W_T$, and $W_H$ denote the group, tail, and entropy window sizes, respectively. To keep the statistics well-defined in the early steps, we use effective window lengths $W_G(t) = \min(W_G, t)$, $W_T(t) = \min(W_T, t)$, and $W_H(t) = \min(W_H, t-1)$. We define grouped confidence

as the moving average over the most recent $W_G(t)$ steps:

$$C_t^G = \frac{1}{W_G(t)} \sum_{s=t-W_G(t)+1}^{t} C_s. \qquad (3)$$

Similarly, tail confidence is defined as

$$C_t^{\text{tail}} = \frac{1}{W_T(t)} \sum_{s=t-W_T(t)+1}^{t} C_s. \qquad (4)$$

For entropy, we compute the historical mean entropy over the past $W_H(t)$ steps:

$$\bar{H}_{t-1} = \frac{1}{W_H(t)} \sum_{s=t-W_H(t)}^{t-1} H_s. \qquad (5)$$

$\Delta H_t \triangleq \bar{H}_t - \bar{H}_{t-1}$ is the entropy increment at step $t$.

**Branch Width: Entropy–Confidence Joint Scheduling.** Once a branching point is triggered, we determine the branch width $B_t$ by jointly considering (i) the local token-distribution entropy and (ii) the grouped confidence across rollouts. At decoding step $t$ for a given prompt $x$, we define the predictive entropy at the current node as $H_t := \mathcal{H}(\pi_\theta(\cdot \mid x, y_{<t}))$. Before applying the scheduling rule, we run a short warm-up phase to collect entropy statistics under the current model and task distribution. This stage does not update model parameters; it only calibrates the entropy scale to avoid over or under-branching caused by prompt or model-dependent entropy magnitudes. During warm-up, we estimate an entropy lower bound $H_{\text{low}}$ and an entropy upper bound $H_{\text{high}}$, where $H_{\text{high}}$ is set as a high-quantile estimate or a moving maximum of the observed entropies. We further define the grouped confidence $C_t^G$ as the aggregation of token confidences over the $G$ rollouts sampled for the same prompt at step $t$. We compute $B_t$ as:

$$B_t = \text{clip}\Big(\text{round}\Big(B_{\min} + \alpha_B \cdot \frac{H_t - H_{\text{low}}}{H_{\text{high}} - H_{\text{low}} + \varepsilon} \\ - \beta_B \cdot \frac{C_t^G - s_{\text{branch}}}{|s_{\text{branch}}| + \varepsilon}\Big), B_{\min}, B_{\max}\Big), \qquad (6)$$

where $B_{\min}$ and $B_{\max}$ denote the minimum and maximum number of branches, respectively. $\alpha_B$ and $\beta_B$ control the relative influence of entropy and confidence, $s_{\text{branch}}$ is a reference confidence level for branching, and $\varepsilon$ is for numerical stability. We first compute a real-valued width, then round it to an integer, and finally clip it to $[B_{\min}, B_{\max}]$. This scheduling rule satisfies: (i) high $H_t$ with low $C_t^G$ increases $B_t$ to encourage exploration; (ii) high $H_t$ with high $C_t^G$ suppresses excessive branching to avoid high-entropy traps; (iii) when $B_t \leq 1$, we treat the node as non-branching and revert to a chain-style rollout at that step.

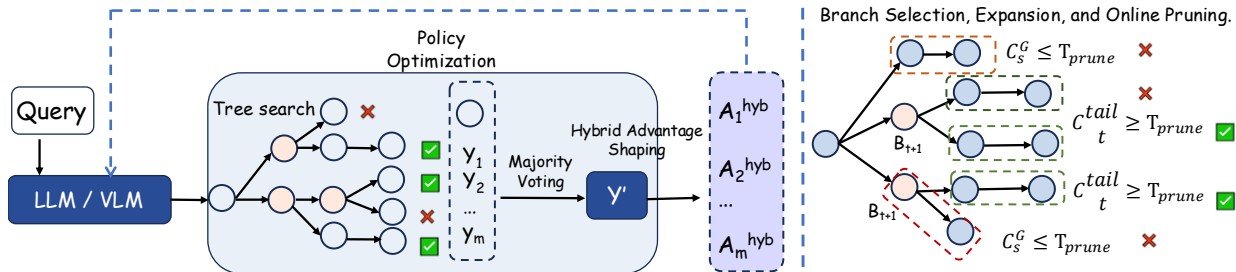

*Figure 3.* Overview of our proposed framework. Given a query, the LLM/VLM performs entropy–confidence guided tree-search rollouts with adaptive branching and confidence-based online pruning. The surviving trajectories produce candidate answers $y_i$, which are aggregated by majority voting to obtain $y'$. We then compute hybrid shaped advantages $A_i^{\text{hyb}}$ for policy optimization, while pruning branches whose grouped confidence falls below $\tau_{\text{prune}}$ to save budget and prevent high-entropy collapse.

**Branch Selection, Expansion, and Online Pruning.** At a branching step $t$, given $B_t$, we select the top-$B_t$ tokens by log-probability under $\pi_\theta(\cdot \mid x, y_{<t})$ as branching candidates. Each child branch is autoregressively expanded until reaching a leaf (a complete answer), yielding candidate trajectories $\{o_i\}_{i=1}^G$. We record the log-probabilities of selected branching tokens for diagnostics. To mitigate rollout collapse and avoid wasting budget on persistently unpromising branches, we perform *online pruning* during rollout: once a branch exhibits sustained low confidence or abnormal uncertainty escalation, we terminate it early so that more budget is allocated to informative branches. We use tail-smoothed confidence $C_{i,t}^{\text{tail}}$ (Eq. 4). *(i) Low-confidence pruning.* We monitor the running minimum of grouped confidence:

$$m_t = \min_{s \leq t} C_s^G. \tag{7}$$

If $m_t < \tau_{\text{prune}}$, we prune the branch at step $t$, where $\tau_{\text{prune}}$ controls the minimum acceptable grouped confidence. *(ii) Tail-decline pruning.* We maintain a counter of consecutive decreases in tail confidence:

$$d_t = \begin{cases} d_{t-1} + 1, & \text{if } C_{i,t}^{\text{tail}} < C_{i,t-1}^{\text{tail}}, \\ 0, & \text{otherwise.} \end{cases} \tag{8}$$

If $d_t \geq S_{\text{tail}}$ and $C_{i,t}^{\text{tail}} \leq \tau_{\text{tail}}$, we prune the branch, where $S_{\text{tail}}$ is the patience length and $\tau_{\text{tail}}$ is the tail-confidence threshold. *(iii) Entropy-spike pruning.* We maintain a consecutive spike counter on entropy increments:

$$r_t = \begin{cases} r_{t-1} + 1, & \text{if } \Delta H_t > \delta_{\text{upper}}, \\ 0, & \text{otherwise.} \end{cases} \tag{9}$$

If $r_t \geq S_\Delta$, we prune the branch (entropy-spike event), where $\delta_{\text{upper}}$ is the spike threshold and $S_\Delta$ is the required consecutive length. Overall, these three criteria provide complementary early-stopping signals: low-confidence pruning removes branches stuck in globally unreliable regions, tail-decline pruning captures trajectories with steadily deteriorating confidence, and entropy-spike pruning filters out branches entering persistently unstable uncertainty regimes. Together, online pruning reallocates rollout budget toward higher-quality trajectories and reduces the risk of high-entropy traps.

**Majority Voting Reward Function.** Following existing TTRL methods (Zuo et al., 2025; Liu et al., 2025), we also employ voting to generate pseudo-labels. For each prompt $x$, we sample $G$ candidate responses $\{o_i\}_{i=1}^G$ under the policy, where $o_i$ denotes the $i$-th response. Let $\text{answer}(o_i)$ be the final answer extracted from $o_i$ and $\mathbb{I}\{\cdot\}$ be the indicator function. We select the majority-voted prediction as the estimated label:

$$\hat{y} = \arg\max_a \sum_{i=1}^G \mathbb{I}\big(\text{answer}(o_i) = a\big). \tag{10}$$

Using $\hat{y}$, we assign a rule-based reward to each candidate trajectory:

$$R_i = \mathbb{I}\big(\text{answer}(o_i) = \hat{y}\big) \in \{0, 1\}. \tag{11}$$

### 3.2. Confidence-Adaptive Clipping

TTRL typically relies on pseudo-rewards that are noisy and biased at the early stage. This *early estimation bias* can repeatedly push the policy toward apparently high-reward neighbors favored by the current sampling distribution, even when these neighbors are not truly reliable. Under a fixed clipping radius, such spurious high-reward trajectories may dominate the update, causing rapid distribution sharpening (entropy drop), premature exploitation, and consequently overfitting with exhausted exploration. To mitigate this issue, we make the trust region *adaptive to confidence*, so that unreliable early signals are prevented from inducing overly aggressive updates. We define the per-token importance ratio at decoding step $t$ as

$$r_{i,t}(\theta) = \frac{\pi_\theta(y_{i,t} \mid x, y_{i,<t})}{\pi_{\text{ref}}(y_{i,t} \mid x, y_{i,<t})}, \ \text{clip}(r_{i,t}, 1 - \epsilon, 1 + \epsilon). \tag{12}$$

To obtain a stable confidence signal along a trajectory, we apply tail smoothing to the token confidence and denote it as

$C_{i,t}^{\text{tail}}$ (Eq. 4). For a trajectory $o_i$ with length $|o_i|$, we define the trajectory-level tail confidence as the average confidence over the last $W_{\text{tail}}$ tokens:

$$C_{\text{tail}}(o_i) = \frac{1}{W_{\text{tail}}} \sum_{t=|o_i|-W_{\text{tail}}+1}^{|o_i|} C_{i,t}^{\text{tail}}, \quad (13)$$

where $W_{\text{tail}}$ is a fixed tail window size (set to 16 in our code). We then adjust the clipping radius using $C_{\text{tail}}(o_i)$:

$$\epsilon(o_i) = \epsilon_{\min} + (\epsilon_{\max} - \epsilon_{\min}) \, \sigma(\kappa(1 - C_{\text{tail}}(o_i))). \quad (14)$$

where $\sigma(z) = \frac{1}{1+e^{-z}}$ is the sigmoid function, $\epsilon_{\min}$ and $\epsilon_{\max}$ specify the minimum/maximum trust region, $\kappa$ controls sensitivity. Intuitively, when $C_{\text{tail}}(o_i)$ is high, $\epsilon(o_i)$ approaches $\epsilon_{\min}$, yielding a tighter trust region that prevents early-stage spurious trajectories from dominating the update. When $C_{\text{tail}}(o_i)$ is low, $\epsilon(o_i)$ approaches $\epsilon_{\max}$, allowing more exploratory updates and avoiding premature entropy collapse.

### 3.3. Entropy–Confidence Hybrid Advantage Shaping

Confidence-adaptive clipping controls the *magnitude* of policy updates, but it does not specify *where* within a trajectory the model should learn more aggressively. In TTRL, the most informative learning signals often arise from locally uncertain decisions: these tokens may correspond to genuinely ambiguous reasoning steps that require exploration, yet they are also the regions where early-stage pseudo-label noise is most likely to manifest. Therefore, beyond a trajectory-level objective that only ranks outcomes within a group, we further shape token-level learning using an entropy–confidence signal, aligning optimization with robust exploration rather than nearby overfitting. We first compute a group-normalized trajectory-level advantage

$$A_{g,i}^{\text{grp}} = \frac{R_{g,i} - \text{mean}(\{R_{g,j}\}_{j=1}^{G})}{\text{std}(\{R_{g,j}\}_{j=1}^{G}) + \varepsilon}, \quad (15)$$

where $g$ indexes a rollout group consisting of the $G$ sampled trajectories for the same prompt, $R_{g,i}$ is the scalar reward of the $i$-th trajectory in group $g$, and $\varepsilon > 0$ is a small constant for numerical stability. We broadcast this trajectory-level advantage to each token on the same trajectory, i.e., $A_{g,i,t}^{\text{grp}} \equiv A_{g,i}^{\text{grp}}$ for all $t \in \{1, \ldots, |o_i|\}$. We then introduce an entropy–confidence hybrid shaping signal that emphasizes learning on uncertain regions:

$$S_{i,t} = \alpha \, H_{i,t} + \beta \, (1 - C_{i,t}), \quad (16)$$

where $H_{i,t}$ is the token entropy at step $t$ on trajectory $o_i$, $C_{i,t} \in (0, 1]$ is the token-level top-$k$ confidence computed from the logits, and $H_{i,t}$ and $(1 - C_{i,t})$ denote masked-whitened versions of entropy and inverse confidence, respectively. The coefficients $\alpha, \beta$ weight the two components.

Finally, we incorporate $S_{i,t}$ into the token-level advantage via

$$A_{i,t}^{\text{hyb}} = A_{g,i}^{\text{grp}}(1 + aS_{i,t}), \quad (17)$$

where $a > 0$ is a scaling factor. This hybrid shaping biases policy updates toward informative yet uncertain decisions, complementing confidence-adaptive clipping to reduce early-stage overfitting and maintain exploration.

### 3.4. Overall Objective

Combining confidence-adaptive clipping and entropy–confidence hybrid advantages, we obtain the ECHO objective:

$$\mathcal{L}_{\text{ECHO}}(\theta) = \mathbb{E}_{x,\{o_i\}} \Bigg[ \frac{1}{G} \sum_{i=1}^{G} \frac{1}{|o_i|} \sum_{t=1}^{|o_i|} \Big( \min \Big( r_{i,t}(\theta) \, A_{i,t}^{\text{hyb}}, $$
$$\text{clip}\big(r_{i,t}(\theta), \, 1 - \epsilon(o_i), \, 1 + \epsilon(o_i)\big) \, A_{i,t}^{\text{hyb}} \Big) $$
$$- \beta_{\text{KL}} \, D_{\text{KL}}\Big( \pi_\theta(\cdot \mid x, y_{i,<t}) \,\big\|\, \pi_{\text{ref}}(\cdot \mid x, y_{i,<t}) \Big) \Big) \Bigg]. $$
$$(18)$$

The first term is the clipped policy-gradient objective with a confidence-adaptive trust region and hybrid advantages, while the KL term constrains policy drift for stability. The pseudocode of ECHO is presented in Algorithm 1 . Following standard GRPO derivations with clipped surrogates, the token-wise gradient of our ECHO objective can be written in a PPO-style gated form:

$$\nabla_\theta \, \ell_{i,t}(\theta) = F_{i,t}^{\text{clip}}(\theta) \, A_{i,t}^{\text{hyb}} \, r_{i,t}(\theta) $$
$$\nabla_\theta \log \pi_\theta(y_{i,t} \mid x_i, y_{i,<t}) \,, \quad (19)$$

where $F_{i,t}^{\text{clip}}(\theta) \in \{0, 1\}$ is the clipping gate induced by the importance ratio $r_{i,t}(\theta)$ and the confidence-adaptive trust-region radius $\epsilon(o_i)$. This view reveals that ECHO alleviates early self-reinforcing overfitting by shrinking $\epsilon(o_i)$ for overconfident trajectories, thereby upper-bounding the impact of spurious high-reward samples under noisy pseudo labels. Meanwhile, the hybrid shaping term $A_{i,t}^{\text{hyb}}$ reallocates gradient mass toward informative yet low-certainty decisions while suppressing degenerate high-entropy traps, mitigating rollout collapse. Finally, the KL regularizer provides a per-prefix pull-back to $\pi_{\text{ref}}$, controlling policy drift and stabilizing exploration. Appendix C provides a detailed derivation and explanation of the ECHO gradient.

## 4. Experiments

### 4.1. Experimental Setup

**Benchmarks**. We train and evaluate our models on two categories of benchmarks: natural-language mathematical

*Table 1.* Overall performance on 5 challenging reasoning tasks. The best result in each block is in **bold**, and the best baseline is underlined. Each cell reports pass@16 accuracy.

| Methods | Training data | AIME2024 | AMC | MATH-500 | GPQA | AIME2025 | Avg. |
|---|---|---|---|---|---|---|---|
| ***Qwen2.5-7B*** | - | 20.0 | 68.7 | 83.0 | 40.4 | 23.3 | 47.1 |
| w/TTRL | | 23.3 | 69.9 | 83.4 | 43.4 | 30.0 | 50.0 |
| w/ETMR | | 24.6 | 71.1 | 84.4 | 42.9 | 30.0 | 50.6 |
| w/EVOL-RL | AIME2024 | 26.7 | 72.3 | 83.6 | 44.4 | 26.7 | 50.7 |
| w/INTUITOR | | 26.7 | 73.5 | 84.4 | 43.5 | 30.0 | 51.6 |
| Ours | | **30.0** | **75.9** | **89.4** | **47.7** | **33.3** | **55.3** |
| Δ | - | 3.3 | 2.4 | 5.0 | 3.3 | 3.3 | 3.7 |
| w/TTRL | | 23.6 | 73.5 | 87.2 | 42.4 | 30.0 | 51.3 |
| w/ETMR | | 26.7 | 72.3 | 88.6 | 44.4 | 33.3 | 53.1 |
| w/EVOL-RL | MATH-500 | 26.7 | 72.3 | 86.0 | 42.9 | 33.3 | 52.2 |
| w/INTUITOR | | 30.3 | 73.5 | 88.0 | 46.0 | 40.0 | 55.6 |
| Ours | | **33.3** | **75.9** | **90.0** | **49.0** | **43.3** | **58.3** |
| Δ | - | 3.0 | 2.4 | 1.4 | 3.0 | 3.3 | 2.7 |
| ***Qwen3-8B*** | - | 50.0 | 80.7 | 91.6 | 70.2 | 50.0 | 68.5 |
| w/TTRL | | 50.0 | 81.9 | 91.8 | 74.7 | 33.3 | 66.3 |
| w/ETMR | | 50.0 | 83.1 | 90.8 | 76.3 | 36.7 | 67.4 |
| w/EVOL-RL | AIME2024 | 46.7 | 85.5 | 94.2 | 78.3 | 34.7 | 67.9 |
| w/INTUITOR | | 50.7 | 88.0 | 94.8 | 79.3 | 56.7 | 73.9 |
| Ours | | **53.3** | **90.4** | **95.4** | **81.3** | **60.0** | **76.1** |
| Δ | - | 2.6 | 2.4 | 0.6 | 2.0 | 3.3 | 2.2 |
| w/TTRL | | 43.3 | 79.5 | 91.8 | 84.3 | 50.0 | 69.1 |
| w/ETMR | | 43.3 | 80.7 | 92.0 | 84.8 | 53.5 | 70.9 |
| w/EVOL-RL | MATH-500 | 36.7 | 83.1 | 92.2 | 85.4 | 46.7 | 68.8 |
| w/INTUITOR | | 43.3 | 85.5 | 93.6 | 87.4 | 53.3 | 72.6 |
| Ours | | **46.7** | **90.4** | **94.6** | **89.9** | **56.7** | **75.6** |
| Δ | - | 3.4 | 4.9 | 1.0 | 2.5 | 3.2 | 3.0 |

reasoning and multimodal mathematical reasoning, and assess both performance and generalization on the corresponding test sets. Specifically, the natural-language benchmarks include AIME2024, AMC, MATH-500, GPQA-Diamond, and AIME2025, covering a broad range of problem types such as algebra, number theory, combinatorics, geometry, and statistics. The multi-modal benchmarks include Geometry3k, GeoQA, MathVision, MathVista, MathVerse, and LogicVista, where each problem typically contains both an image and text, spanning diverse visual reasoning scenarios such as geometry, charts, and tables. Detailed dataset descriptions are provided in the Appendix A.1.

**Baselines**. We compare against the following representative methods: TTRL (Zuo et al., 2025), ETMR (Liu et al., 2025), EVOL-RL (Zhou et al., 2025), INTUITOR (Zhao et al., 2025a) , and MM-UPT (Wei et al., 2025), which cover TTRL methods (TTRL, ETMR), evolutionary unsupervised reinforcement learning (EVOL-RL), internal-feedback-driven self-improvement (INTUITOR), and a multi-modal unsupervised post-training framework (MM-UPT). We use pass@n as the unified evaluation metric for all methods, enabling

a comprehensive assessment of our approach in both pure-text and vision–language mathematical reasoning settings. Additional Model details and implementation information are included in the Appendix A.2 and Appendix A.3 .

### 4.2. Main Results

Our main results are summarized in Tables 1 and Table 2. Overall, ECHO demonstrates consistent and substantial advantages on both natural-language reasoning benchmarks (pass@16) and multimodal reasoning benchmarks (pass@1), indicating strong transferability and robustness. We highlight three key findings: (1) **ECHO consistently improves pass@16 over strong baselines and remains robust to changes in training data and environments**. As shown in Table 1, ECHO steadily outperforms a range of test-time optimization methods and confidence baselines on both Qwen2.5-7B and Qwen3-8B (Think) backbones, delivering consistent gains across different training setups. In aggregate, ECHO delivers consistent average gains of 0.63%–12.36% across natural-language reasoning benchmarks, and can achieve up to 12.36% improvements on the

*Table 2.* Overall performance on four multimodal reasoning benchmarks. Best results are in **bold**. Each cell reports pass@1 accuracy.

| Methods | Training data | Math Vision | Math Verse | Math Vista | LogicVista | Avg |
|---|---|---|---|---|---|---|
| ***Qwen2.5-VL-7B*** | - | 24.9 | 43.8 | 66.3 | 26.0 | 40.3 |
| w/ MM-UPT | Geometry3k | 27.3 | 42.5 | 68.5 | 26.6 | 41.2 |
| Ours | | **28.1** | **44.5** | **69.0** | **27.9** | **42.4** |
| Δ | - | +0.8 | +0.7 | +0.5 | +1.3 | +1.2 |
| w/ MM-UPT | GeoQA | 27.1 | 43.7 | 68.9 | 26.8 | 41.6 |
| Ours | | **28.5** | **44.7** | **69.6** | **28.4** | **42.8** |
| Δ | - | +1.4 | +1.0 | +0.7 | +1.6 | +1.2 |
| ***Qwen3-VL-8B*** | - | 62.7 | 62.1 | 77.2 | 22.7 | 56.2 |
| w/ MM-UPT | Geometry3k | 64.7 | 64.1 | 79.2 | 23.9 | 58.0 |
| Ours | | **65.4** | **65.7** | **79.8** | **25.9** | **59.2** |
| Δ | - | +0.7 | +1.6 | +1.6 | +2.0 | +1.5 |
| w/ MM-UPT | GeoQA | 64.1 | 64.6 | 79.5 | 24.2 | 58.1 |
| Ours | | **66.5** | **67.3** | **80.9** | **27.3** | **60.5** |
| Δ | - | +2.4 | +2.7 | +1.4 | +3.1 | +2.4 |

most challenging tasks (e.g., AIME2025). These results suggest that ECHO more effectively filters low-quality trajectories and mitigates search degeneration across varying training distributions, leading to reliable and sustained improvements in reasoning. (2) **ECHO extends seamlessly to multimodal reasoning and yields stable pass@1 gains.** Table 2 shows that ECHO remains effective on multimodal benchmarks: across VLM backbones such as Qwen2.5-VL-7B and Qwen3-VL-8B, ECHO consistently improves the average pass@1 by about 2.6%–4.1%, and achieves up to 12.8% relative gains on the most challenging multimodal tasks (LogicVista). This indicates that ECHO is not limited to text-only reasoning, but can also enhance cross-modal reasoning by improving trajectory quality and final-answer reliability via entropy–confidence scheduling. (3) **ECHO remains consistently effective even under strict IID settings**. Table 4 further verifies the robustness of ECHO under strict IID evaluation (training and testing on the same dataset). Across multiple benchmarks and model scales, ECHO delivers consistent improvements: it increases average accuracy by 4.01%–8.88% (relative gain in the Avg. column) and achieves up to 16.50% gains on high-difficulty tasks. This suggests that even without relying on distribution shift, ECHO can continuously improve reasoning quality through more effective search and update dynamics. In summary, ECHO yields consistent advantages across model scales, training data, and task modalities, validating its effectiveness in balancing exploration and stability for test-time reasoning optimization and demonstrating strong generalization capability.

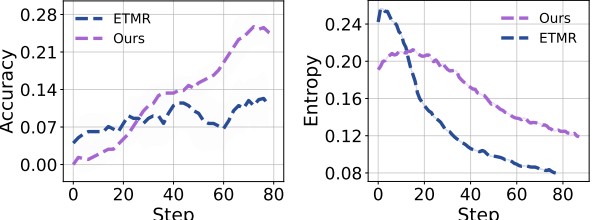

*Figure 4.* Training dynamics for Ours and ETMR. Both models trained on AIME2024.

### 4.3. Ablation Study

Table 3 presents an ablation study of the three key components in ECHO. EC-Tree improves trajectory quality during tree search via joint entropy–confidence scheduling; CA-Clip stabilizes policy updates with confidence-adaptive clipping, mitigating early estimation bias; and E-SC Adv enhances learning from exploratory yet uncertain samples through entropy–confidence advantage shaping. Overall, removing any component degrades performance, while the full ECHO consistently achieves the best results across all benchmarks and both training settings. In particular, removing EC-Tree results in the largest overall drop, indicating that high-quality candidate trajectories underpin performance gains. Removing CA-Clip causes a dramatic decline on challenging benchmarks (e.g., AIME2025 drops from 54.3 to 30.1), highlighting that stable confidence constraints are crucial for suppressing early bias and improving generalization. Finally, removing E-SC Adv yields consistent but milder degradations across multiple benchmarks, validating the sustained contribution of entropy-confidence advantages in promoting exploratory learning.

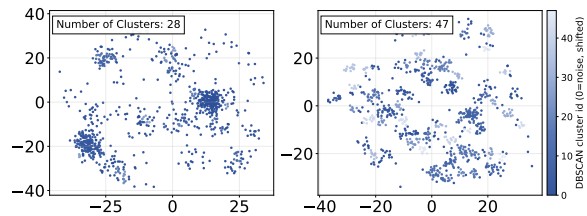

*Figure 5.* Visualization of Rollout diversity: ETMR and Ours.

## 4.4. Quantitative Analysis

**This section examines whether our method mitigates the two challenges raised in Section 1**. Following work (Dong et al., 2025), we reuse the rollout configuration from our main experiments and randomly collect branching trajectories across multiple rollout steps (300 problems and 4.8k trajectories on MATH-500). We then encode trajectories with BGEM3, apply PCA for dimensionality reduction, and use DBSCAN clustering to visualize the trajectory distribution to assess whether the branching budget is evenly allocated or collapses into a small number of dominant paths (Figure). (1) **High-entropy collapse**. As shown in Figure 4 and Figure 5, purely entropy-driven tree search (TTRL/ETMR) tends to exhibit consecutive high-entropy transitions, where the budget is dominated by a few trajectories, resulting in limited coverage and triggering high-entropy rollout collapse. We incorporate confidence constraints and online pruning during branching to prevent over-expansion into noisy high-entropy regions and encourage a more balanced allocation of branches. (2) **Pseudo-label overfitting**. Figure 4 indicates that entropy-driven methods often reduce entropy too rapidly in early stages and amplify unreliable pseudo-labels, leading to premature convergence and self-reinforcing overfitting. We adopt confidence-adaptive clipping and entropy–confidence advantage shaping to slow entropy decay and reduce the dominance of early noisy supervision in policy updates. Overall, our method simultaneously alleviates rollout collapse caused by high-entropy branching and self-reinforcing overfitting induced by early pseudo-label bias, resulting in more robust test-time reinforcement learning behavior.

Due to space constraints, we provide additional experiments and analyses in the appendix. Appendix B.1 investigates scenarios in which ECHO may fail; Appendix B.2 compares label-free ECHO with supervised GRPO; and Appendix B.3 reports performance under varying Pass@$K$ settings.

## 5. Related Work

### 5.1. Reinforcement Learning for LLM Reasoning

Reinforcement learning (Wiering & Van Otterlo, 2012; Kaelbling et al., 1996) plays an important role in enhancing the reasoning capabilities of large language models, especially

under the paradigm of Reinforcement Learning from Human Feedback (RLHF) (Chaudhari et al., 2025). RLHF typically uses a reward model to capture human preferences and applies Proximal Policy Optimization (PPO) (Schulman et al., 2017; Wang et al., 2020) to update the policy with on-policy samples. To improve training stability and sample efficiency, PPO often incorporates a KL constraint and variance-reduction techniques such as Generalized Advantage Estimation, enabling the model to progressively improve generation quality without drifting too far from a reference policy. Recent works (Shao et al., 2024; Yu et al., 2025; Zheng et al., 2025) have further refined the PPO pipeline; for example, GRPO estimates the baseline via within-group score normalization, replacing explicit critic learning and mitigating the associated instability. However, most of these approaches (Hu et al., 2025; Xi et al., 2025) are still optimized primarily on supervised training data, while at test time the model is often required to produce longer and more complex chains of thought on out-of-distribution problems. This mismatch reduces robustness and amplifies errors, limiting real-world performance.

### 5.2. Self-Improvement and Test-time Reinforcement Learning

Methods for improving LLM reasoning are typically trained with ground-truth labels or verifiable external feedback. When such supervision is unavailable or the distribution shifts, reward signals can become unreliable and sparse, limiting optimization. To mitigate this, recent work constructs rewards without ground-truth labels (Zuo et al., 2025; Zanella et al., 2025), mainly in two ways: (1) confidence-based self-rewarding, which derives rewards from the model's output statistics (e.g., low entropy/high confidence) and consistency measures such as agreement across samples and stability under paraphrases (Zhao et al., 2025a; Yuan et al., 2024; Prabhudesai et al., 2025; Shafayat et al., 2025); and (2) majority-bootstrapped pseudo-label rewards, which sample multiple outputs and treat the majority answer as a pseudo-label for RL-style updates (e.g., TTRL) (Zuo et al., 2025). Despite their effectiveness, these methods have limitations: TTRL's chain-based rollouts often require heavy sampling to obtain reliable pseudo-labels, and ETMR (Liu et al., 2025) improves exploration with entropy-fork trees but can still suffer from entropy collapse, where high-entropy branches dominate and destabilize learning.

## 6. Conclusion

This work proposes ECHO, an entropy–confidence hybrid test-time RL framework for self-improvement without external supervision. ECHO combines entropy–confidence guided tree-structured rollouts with online pruning to focus computation on informative branches and avoid

high-entropy traps. It further stabilizes optimization via confidence-adaptive clipping and entropy–confidence hybrid advantage shaping, prioritizing learning on uncertain yet decision-critical tokens. Extensive experiments validate its effectiveness.

## Acknowledgements

This work is partially supported by the National Natural Science Foundation of China under Grant No. 62576083. We extend special thanks to the National Supercomputer Center in Guangzhou for their computational support.

## Impact Statement

This paper presents work whose goal is to advance the field of machine learning. There are many potential societal consequences of our work, none of which we feel must be specifically highlighted here.

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

*Table 3.* Performance of Qwen2.5-7B-Base with ECHO
and its ablations on five benchmarks. Each cell reports pass@16 accuracy.

| Models | Datasets | AIME2024 | AMC | MATH-500 | GPQA | AIME2025 |
|--------|----------|----------|-----|----------|------|----------|
| ***Qwen2.5-7B-Base*** | – | 20.0 | 68.7 | 83.0 | 40.4 | 27.3 |
| -EC-Tree | | 23.3 | 54.2 | 74.6 | 32.3 | 29.7 |
| -CA-Clip | **AIME2024** | 26.7 | 69.9 | 84.0 | 43.9 | 30.1 |
| -E-SC Adv | | 26.7 | 78.7 | 84.0 | 44.4 | 53.5 |
| **Ours** | | **30.0** | **71.1** | **89.4** | **47.7** | **54.3** |
| -EC-Tree | | 23.3 | 56.6 | 77.4 | 39.4 | 33.5 |
| -CA-Clip | **MATH-500** | 30.0 | 72.3 | 88.2 | 46.5 | 41.0 |
| -E-SC Adv | | 30.0 | 73.5 | 89.4 | 46.5 | 42.5 |
| **Ours** | | **33.3** | **75.9** | **90.0** | **49.0** | **43.3** |

# A. Experimental Setup

## A.1. Benchmarks

We evaluate our model on both text-only and multimodal mathematical reasoning benchmarks. Details is shown as follows.

**(1) Text-only mathematical reasoning benchmarks.** We evaluate the model on AIME2024, AMC, MATH-500, GPQA-Diamond, and AIME2025 to assess its robustness under challenging reasoning and cross-domain knowledge requirements. **AIME2024 (Li et al., 2024) / AIME2025.** These benchmarks consist of the AIME 2024/2025 I & II exams (30 problems each).They represent contest-level mathematical reasoning tasks where answers are typically short (often integers). **AMC (Li et al., 2024)** is a classic math competition benchmark with relatively short problems covering algebra, geometry, combinatorics, and number theory. It is commonly used to evaluate fundamental mathematical reasoning and robustness (grading can be based on option matching or numeric answers, depending on the evaluation setup). **MATH-500 (Hendrycks et al., 2021)** is a 500-problem subset sampled from the MATH test set, covering diverse mathematical topics. Answers are often exact strings or mathematical expressions; evaluation typically involves answer normalization and optionally symbolic-equivalence checking to reduce formatting-induced mismatches. **GPQA-Diamond (?)** is a graduate-level, Google-proof science QA benchmark. Its Diamond subset contains 198 high-consistency questions, primarily in multiple-choice format, spanning Biology, Chemistry, and Physics. It emphasizes domain knowledge and complex reasoning.

**(2) Multimodal mathematical reasoning benchmarks.** We further evaluate visual mathematical reasoning on MathVision, MathVista, MathVerse, and LogicVista. **MathVision (Wang et al., 2024).** It contains 3,040 competition-style math problems with visual context, covering 12 grade levels, 16 subjects, and 5 difficulty levels, including topics such as analytic geometry, combinatorial geometry, and topology. **MathVista (Lu et al., 2023).** It includes 1,000 problems spanning diverse visual scenarios such as geometry, charts, and tables, and is designed to evaluate mathematical reasoning in visual contexts. **MathVerse (Zhang et al., 2024).** Its test set contains 3,940 diagram-based multi-subject math problems collected from public sources, focusing on core visual reasoning scenarios such as plane and solid geometry, enabling fair and fine-grained evaluation.

## A.2. Models Details

We conduct experiments across backbone models of different scales and systematically compare a wide range of baseline methods. Our backbones mainly include the **Qwen** family (Qwen2.5-Math-1.5B (Team et al., 2024), Qwen2.5-Math-7B (Team et al., 2024), Qwen2.5-7B-Base (Hui et al., 2024), and Qwen3-8B-Base (Yang et al., 2025)), as well as **Qwen multimodal** models (Qwen2.5-VL-7B (Bai et al., 2025b) and Qwen3-VL-8B (Bai et al., 2025a)). We also validate our findings on the **LLaMA** family (LLaMA-3.1-8B-Instruct (Grattafiori et al., 2024)). The baselines cover representative test-time and post-training strategies, including TTRL, ETMR, EVOL-RL, INTUITOR, and the multimodal test-time reinforcement learning method MM-UPT. We briefly describe these baselines below:

*Table 4.* Overall performance on 5 challenging reasoning tasks. The best result is in **bold**, and the best baseline is underlined. Each cell reports pass@16 accuracy (averaged over 32 rollouts).

| Methods & datasets | AIME2024 | AMC | MATH-500 | GPQA | AIME2025 | AVG. |
|---|---|---|---|---|---|---|
| *Qwen2.5-Math-1.5B* | 13.3 | 28.9 | 71.0 | 30.8 | 20.0 | 32.8 |
| w/TTRL | 15.8 | 30.1 | 73.0 | 34.8 | 20.0 | 34.7 |
| w/ETMR | 16.7 | 32.5 | 76.9 | 35.4 | 23.3 | 37.0 |
| w/INTUITOR | 20.0 | 34.9 | 77.4 | 35.9 | 23.3 | 38.3 |
| **Ours** | **23.3** | **39.8** | **80.4** | **38.4** | **26.7** | **41.7** |
| Δ | 3.3 | 4.9 | 3.0 | 2.5 | 3.4 | 3.4 |
| *Qwen2.5-Math-7B* | 36.7 | 69.9 | 85.0 | 60.1 | 30.0 | 56.3 |
| w/TTRL | 40.0 | 73.5 | 86.4 | 60.6 | 36.7 | 59.4 |
| w/ETMR | 40.0 | 73.5 | 88.4 | 61.6 | 40.0 | 60.7 |
| w/INTUITOR | 43.3 | 75.4 | 90.0 | 62.6 | 40.0 | 62.3 |
| **Ours** | **46.7** | **77.1** | **92.4** | **64.1** | **43.3** | **64.7** |
| Δ | 3.4 | 1.7 | 2.4 | 1.5 | 3.3 | 2.5 |
| *Qwen2.5-7B* | 20.0 | 68.7 | 83.0 | 55.6 | 23.3 | 50.1 |
| w/TTRL | 23.3 | 71.1 | 87.2 | 58.1 | 30.0 | 53.9 |
| w/ETMR | 24.6 | 72.3 | 88.6 | 60.6 | 30.0 | 55.2 |
| w/INTUITOR | 26.7 | 73.5 | 88.0 | 61.1 | 30.0 | 55.9 |
| **Ours** | **30.0** | **75.9** | **90.0** | **62.6** | **33.3** | **58.4** |
| Δ | 3.3 | 2.4 | 1.4 | 1.5 | 3.3 | 2.5 |
| *Llama3-8B* | 20.0 | 65.1 | 82.0 | 52.0 | 23.3 | 48.5 |
| w/TTRL | 16.7 | 66.3 | 84.0 | 57.1 | 23.3 | 49.5 |
| w/INTUITOR | 20.0 | 66.3 | 86.2 | 58.6 | 26.7 | 51.6 |
| w/ETMR | 20.0 | 67.5 | 86.4 | 58.6 | 26.7 | 51.8 |
| **Ours** | **23.3** | **68.7** | **88.8** | **60.6** | **30.0** | **54.3** |
| Δ | 3.3 | 1.2 | 2.4 | 2.0 | 3.3 | 2.4 |

- **TTRL** (Zuo et al., 2025) is a novel approach that trains LLMs with reinforcement learning (RL) on unlabeled data. It leverages priors encoded in pre-trained models as learning signals, enabling self-evolution and self-improvement at test time.

- **ETMR** (Liu et al., 2025) extends TTRL by replacing the chain-based rollout with an *Entropy-fork Tree Majority Rollout*, aiming to better balance exploration and exploitation in test-time reinforcement learning and improve the effectiveness of search and sampling.

- **EVOL-RL** (Zhou et al., 2025) introduces a self-evolution framework that jointly considers stability and diversity: it retains the stability of majority-vote selection while incorporating an explicit *variation incentive* that rewards semantic novelty, thereby mitigating the *entropy collapse* issue that can arise in TTRL.

- **INTUITOR** (Zhao et al., 2025a) is an RLIF method whose sole reward signal is the model's own confidence, termed *confidence*. It optimizes the model using this intrinsic signal, without requiring external labels or additional evaluators.

- **MM-UPT** (Wei et al., 2025) targets unlabeled multimodal RL by generating pseudo-labels via a voting mechanism, providing supervision signals to align and optimize multimodal models without human annotations.

### A.3. Implementation Details

Following the setups of recent related work (Zuo et al., 2025), we apply GRPO on each benchmark to implement ECHO. Specifically, during the rollout stage, for each problem prompt, the policy model first generates 64 candidate responses, which are used for voting-based pseudo-label construction. We then downsample 32 responses per prompt from these

candidates for training updates. During generation, we set the maximum response length to 3,072 tokens to balance reasoning completeness and computational cost. The general hyperparameters are summarized in Table 5. The hyperparameters and configurations specific to our ECHO method are provided in Table 6.

*Table 5.* Overall hyperparameters for label-free training, following the TTRL setup.

| Hyperparameter | Value |
|---|---|
| Train Batch Size | 5 |
| PPO Mini-Batch Size | 1 |
| PPO Micro-Batch Size | 1 |
| Rollouts for Majority Vote | 64 |
| Rollouts Used for Training | 32 |
| Validation Temperature | 0.6 |
| Learning Rate | 5e-7 |
| Use KL Loss | True |
| KL Loss Coefficient | 0.001 |

*Table 6.* Key hyperparameter settings for ECHO.

| Key Hyperparameter | Value |
|---|---|
| Entropy high $H_{\text{high}}$ | 3.5 |
| Entropy low $H_{\text{low}}$ | 1.0 |
| Branch threshold $s_{\text{branch}}$ | 1.2 |
| Min branches $B_{\text{min}}$ | 1 |
| Max branches $B_{\text{max}}$ | 4 |
| Prune threshold $\tau_{\text{prune}}$ | 0.4 |
| Tail window $W_T$ | 8 |
| Group entropy window $W_H$ | 4 |
| Tail confidence threshold $\tau_{\text{tail}}$ | 1 |
| Consecutive length $S_\Delta$ | 3 |
| Patience length $S_{tail}$ | 3 |
| Tail-confidence threshold $\tau_{tail}$ | 1 |
| Spike threshold $\delta_{upper}$ | 0.5 |

# B. Additional Results and Analysis

### B.1. When Might ECHO Fail?

**Inappropriate pruning hyperparameters.** Tree-structured rollouts are highly sensitive to pruning-related hyperparameters. When a problem requires a long intermediate derivation phase with *low confidence*, overly relying on short-window confidence statistics can **prematurely terminate** trajectories that would otherwise reach the correct answer. This issue is particularly severe under delayed outcome signals, where good and bad trajectories are mainly distinguishable only at the final answer, thereby limiting the model performance. The results in Tables 7 and Table 8 provide direct evidence. For AIME2024 (Table 7), when the pruning threshold is increased from $\tau_{\text{prune}} = 0.4$ to a more aggressive setting of 1.2, **both Pass@1 and Pass@2 drop to 0**, and Pass@16 decreases from **30.0** to **16.7**, indicating that many potentially correct trajectories are pruned before reaching the end. Meanwhile, overly small or improper thresholds can also hurt stability: for example, at $\tau_{\text{prune}} = 0.2$, Pass@16 is only **16.7**, substantially lower than the **30.0** achieved at $\tau_{\text{prune}} = 0.4$, suggesting that a careful balance between pruning strength and noise tolerance is required. Similarly, on AMC (Table 8), the tail-window size $W_T$ has a significant impact. Increasing $W_T$ from 2 to 8 improves Pass@16 from **60.2** to **75.9**, and Pass@8 from **55.4** to **63.9**. This indicates that a longer window better smooths confidence fluctuations during intermediate reasoning, reducing *false pruning* caused by transient confidence drops and increasing the probability of reaching the correct final answer. In contrast, overly short windows are more susceptible to local noise, making pruning decisions unstable and degrading Pass@K. Overall, $\tau_{\text{prune}}$ (pruning aggressiveness) and $W_T$ (confidence smoothing scale) are the two key hyperparameters

*Table 7.* Pass@K under different pruning thresholds $\tau_{\text{prune}}$. We conduct this experiment on AIME2024 dataset without labels.

| $\tau_{\text{prune}}$ | Pass@1 | Pass@2 | Pass@4 | Pass@8 | Pass@16 |
|---|---|---|---|---|---|
| 0.2 | 6.7 | 10.0 | 10.0 | 13.3 | 16.7 |
| 0.4 | 10 | 10.0 | 13.7 | 16.6 | 30.0 |
| 0.8 | 6.7 | 11.6 | 13.7 | 13.3 | 26.7 |
| 1.2 | 0.0 | 0.0 | 6.7 | 10.0 | 16.7 |

*Table 8.* Pass@K under different tail-window sizes $W_T$ (fix $\tau_{\text{prune}} = 0.04$). We conduct this experiment on AMC dataset without labels.

| $W_T$ | Pass@1 | Pass@2 | Pass@4 | Pass@8 | Pass@16 |
|---|---|---|---|---|---|
| 2 | 20.5 | 21.7 | 55.4 | 55.4 | 60.2 |
| 4 | 22.9 | 27.7 | 61.4 | 61.4 | 66.3 |
| 8 | 22.9 | 28.9 | 62.7 | 63.9 | 75.9 |
| 10 | 28.9 | 28.9 | 63.9 | 64.1 | 75.9 |

governing this failure mode: **overly aggressive thresholds or overly short windows amplify false pruning, thereby weakening or even offsetting the benefits of our method**.

**Lack of Prior Knowledge on the Target Task.** As a label-free, self-bootstrapped optimization framework, ECHO derives its training signal from majority voting over the model's own rollouts. This implicitly requires that the backbone model can produce a non-trivial fraction of correct candidates on the target task, so that the voted pseudo-labels are positively correlated with true correctness. Otherwise, majority voting is more likely to converge to an incorrect consensus, pushing updates toward noisy signals and yielding diminished gains or even negative transfer. This issue becomes particularly salient when the training data distribution is mismatched with the target benchmarks. Tables 9 and Table 10 illustrate this failure mode. For training on ReClor in Table 9, ReClor mainly consists of reading comprehension and logical multiple-choice questions, whose signals emphasize linguistic logic and discriminative reasoning. When transferring to AIME/AMC/MATH-500/GPQA, which demand competition-level mathematical reasoning and graduate-level science QA, the transferable math/science priors are limited. With the weaker Qwen2.5-7B-Base backbone, our method yields only marginal improvements and can even degrade performance (e.g., MATH-500 drops from 83.0 to 81.9), suggesting that the voted pseudo-labels are dominated by incorrect trajectories and the updates fail to align with the target capabilities. In contrast, the stronger Qwen3-8B-Base backbone provides richer task-relevant priors and thus more correct candidates in rollouts, leading to more stable learning signals and more reliable overall behavior.

A similar phenomenon appears in the multimodal setting in Table 10. ThinkLite-11K can be viewed as a dataset biased toward lightweight reasoning alignment, which differs substantially from visual-math benchmarks such as MathVision, MathVerse, MathVista, and LogicVista in problem structure, required knowledge, and long-horizon reasoning patterns. Under such distribution mismatch, even with a stronger Qwen3-VL-8B-Instruct backbone, ECHO may still incur drops on certain datasets (e.g., MathVision and MathVista), indicating that when key priors (e.g., geometry, chart/table understanding, and multi-step derivations) are missing, pseudo-labels from voting become less reliable and negative transfer becomes more likely.

In summary, both ReClor and ThinkLite-11K deviate notably from the target evaluation benchmarks in task format and required skill dimensions. When the backbone lacks task-specific priors, the vote-based pseudo-label to self-bootstrapped update pipeline of ECHO can be dominated by systematic noise, resulting in limited gains or even failures.

### B.2. Comparing label-free ECHO with supervised GRPO (RLVR)

We compare ECHO, a label-free test-time reinforcement learning method, against supervised GRPO in the RLVR setting. Since GRPO optimizes with explicit supervised signals while ECHO relies only on self-generated rollouts, this study provides a head-to-head comparison between supervised and label-free optimization under comparable inference-time budgets.

*Table 9.* Performance comparison with different base models and training datasets (on ReClor). Each cell reports pass@16 accuracy.

| Models | Training dataset | AIME2024 | AMC | MATH-500 | GPQA |
|---|---|---|---|---|---|
| *Qwen2.5-7B-Base* | - | 20.0 | 68.7 | 83.0 | 40.4 |
| TTRL | ReClor | 16.7 | 69.7 | 80.7 | 41.4 |
| Ours | | 20.0 | 71.1 | 81.9 | 41.9 |
| *Qwen3-8B-Base* | - | 40.0 | 80.7 | 91.6 | 70.2 |
| TTRL | ReClor | 36.7 | 78.1 | 90.5 | 72.7 |
| Ours | | 40.0 | 79.5 | 89.2 | 69.2 |

*Table 10.* Performance comparison with different base models and training datasets (on ThinkLite-11k). Each cell reports pass@1 accuracy.

| Models | Training dataset | Math Vision | Math Verse | Math Vista | LogicVista |
|---|---|---|---|---|---|
| *Qwen2.5-VL-7B* | | 24.9 | 43.8 | 66.3 | 26.0 |
| MM-UPT | ThinkLite-11K | 22.6 | 38.5 | 60.7 | 22.7 |
| OURS | | 24.6 | 44.8 | 67.3 | 25.3 |
| *Qwen3-VL-8B* | | 62.7 | 62.1 | 77.2 | 22.7 |
| MM-UPT | ThinkLite-11K | 60.7 | 60.5 | 75.6 | 19.8 |
| OURS | | 62.0 | 63.1 | 72.3 | 22.5 |

**Text-only reasoning.** As shown in Table 11, on the Qwen3-8B-Base backbone, ECHO is close to supervised GRPO on most benchmarks: it is lower by 1.9 on AIME2024, lower by 0.8 on AMC, higher by 2.0 on MATH-500, and lower by 0.8 on GPQA. These results indicate that, without external labels, ECHO can match supervised RLVR on three benchmarks within one point and even outperform it on MATH-500.

**Vision-based reasoning.** Table 12 reports multimodal results on Qwen2.5-VL-7B. Compared with supervised GRPO, ECHO is lower by 0.2 on MathVision and lower by 0.3 on MathVista, while the gap is larger on MathVerse and LogicVista (lower by 1.9 and 2.1, respectively). This pattern suggests that label-free optimization remains more challenging on vision tasks that require stronger visual grounding and more reliable credit assignment.

**Summary.** Overall, Tables 11 and 12 show that ECHO can be competitive with supervised GRPO, matching it closely on several benchmarks and surpassing it on MATH-500, while the remaining differences are more pronounced on vision-heavy tasks.

### B.3. Performance under varying Pass@K

We further evaluate how model performance changes as the sampling budget increases by reporting results under different Pass@K settings. Figure 6 and Figure 7 summarize the trends on both text-only and vision-based reasoning benchmarks, which helps assess the stability of inference-time search and the robustness of cross-dataset generalization.

**Text-only reasoning benchmarks.** In Figure 6, we train Qwen2.5-7B-Base on AIME2024 and evaluate it on AIME2024, AMC, GPQA, and MATH-500. Across all datasets, accuracy consistently increases with larger $K$, indicating that additional rollouts improve the chance of discovering a correct solution for hard reasoning problems. The gains are most pronounced when increasing $K$ from 1 to 4, and then gradually saturate as $K$ becomes larger. Overall, our method maintains strong performance across the full range of $K$ and typically achieves higher accuracy at medium and large $K$, suggesting that it benefits more effectively from increased sampling budgets and yields more stable improvements when additional candidates are available.

**Vision-based reasoning benchmarks.** Figure 7 reports the same analysis for Qwen2.5-VL-7B trained on Geometry3k and evaluated on LogicVista, MathVerse, MathVision, and MathVista. Similar to the text-only setting, performance improves monotonically with $K$ on all four benchmarks, showing that multi-sample decoding is also beneficial for visual reasoning. The performance gains are again concentrated in the low-$K$ regime, while improvements become smaller at higher $K$. Our

*Table 11.* Performance comparison on four benchmarks.

| Models | AIME2024 | AMC | MATH-500 | GPQA |
|---|---|---|---|---|
| *Qwen3-8B-Base* | 39.4 | 77.6 | 91.5 | 70.0 |
| GRPO | 55.2 | 91.2 | 93.4 | 82.1 |
| Ours | 53.3 | 90.4 | 95.4 | 81.3 |

*Table 12.* Performance comparison on four vision-reasoning benchmarks.

| Models | Math Vision | Math Verse | Math Vista | LogicVista |
|---|---|---|---|---|
| *Qwen2.5-VL-7B* | 24.9 | 43.8 | 66.3 | 26.0 |
| GRPO | 28.3 | 46.4 | 69.3 | 30.0 |
| Ours | 28.1 | 44.5 | 69.0 | 27.9 |

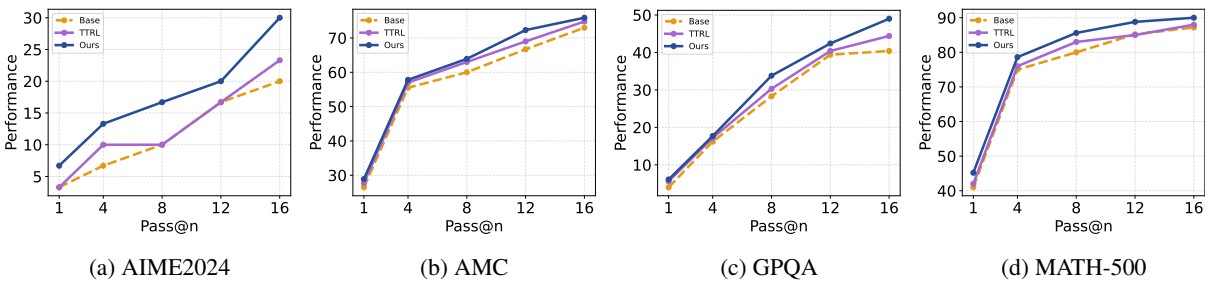

| (a) AIME2024 | (b) AMC | (c) GPQA | (d) MATH-500 |
|---|---|---|---|

*Figure 6.* We train Qwen2.5-7B-Base on AIME2024 and evaluate it on other benchmarks, reporting performance under different Pass@K settings to assess cross-dataset generalization and reasoning stability.

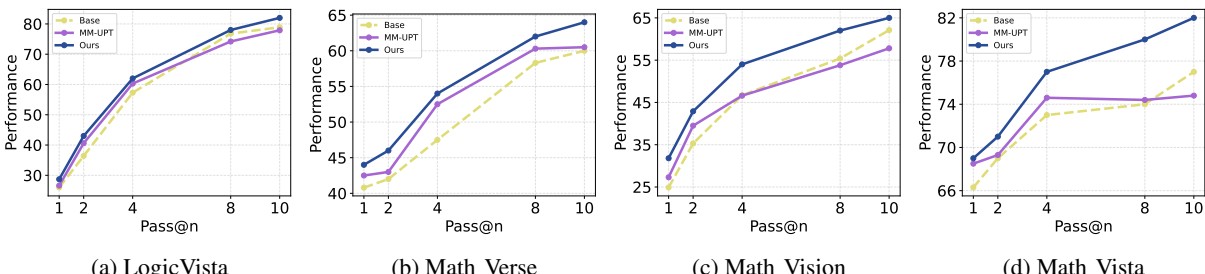

| (a) LogicVista | (b) Math_Verse | (c) Math_Vision | (d) Math_Vista |
|---|---|---|---|

*Figure 7.* We train Qwen2.5-VL-7B on Geometry3k and evaluate it on other benchmarks, reporting performance under different Pass@K settings to assess cross-dataset generalization and reasoning stability.

method remains competitive across datasets and demonstrates stronger utilization of increased sampling budgets, indicating that it can better convert additional rollouts into effective reasoning improvements under cross-dataset transfer.

## C. Detailed theoretical derivations and explanations

### C.1. Notation and Objective

For each prompt $x_i$, we sample a trajectory $o_i = (y_{i,1}, \ldots, y_{i,|o_i|})$. Define the per-token importance ratio

$$r_{i,t}(\theta) = \frac{\pi_\theta(y_{i,t} \mid x_i, y_{i,<t})}{\pi_{\text{ref}}(y_{i,t} \mid x_i, y_{i,<t})}, \qquad \epsilon_i = \epsilon(o_i). \tag{20}$$

Let $A_{i,t}^{\text{hyb}}$ denote the hybrid advantage defined in the main text. During policy optimization, we treat $A_{i,t}^{\text{hyb}}$ and $\epsilon_i$ as fixed quantities (i.e., detached from backpropagation). The per-token objective of ECHO is

$$J_{i,t}(\theta) = \min\Big(r_{i,t}(\theta)\, A_{i,t}^{\text{hyb}},\ \text{clip}\big(r_{i,t}(\theta), 1 - \epsilon_i, 1 + \epsilon_i\big)\, A_{i,t}^{\text{hyb}}\Big) - \beta_{\text{KL}}\, D_{i,t}^{\text{KL}}(\theta), \tag{21}$$

where

$$D_{i,t}^{\text{KL}}(\theta) = D_{\text{KL}}\Big(\pi_\theta(\cdot \mid x_i, y_{i,<t}) \,\big\|\, \pi_{\text{ref}}(\cdot \mid x_i, y_{i,<t})\Big). \tag{22}$$

## C.2. Piecewise Form and Clipping Gate for $\min(\cdot)$

Let $A = A_{i,t}^{\text{hyb}}$, $r = r_{i,t}(\theta)$, $\epsilon = \epsilon_i$, and $\bar{r} = \text{clip}(r, 1 - \epsilon, 1 + \epsilon)$. Define

$$s(r) = \min(rA, \bar{r}A). \tag{23}$$

**Lemma C.1** (Piecewise surrogate). *The function $s(r)$ admits the following piecewise form:*

$$s(r) = \begin{cases} (1 + \epsilon)A, & A > 0,\ r > 1 + \epsilon, \\ (1 - \epsilon)A, & A < 0,\ r < 1 - \epsilon, \\ rA, & \text{otherwise.} \end{cases} \tag{24}$$

*Proof.* The statement follows by separately considering the cases $A > 0$ and $A < 0$ and determining which branch is selected by $\min(\cdot)$. $\qquad\square$

## C.3. Gradient of the Importance Ratio

**Lemma C.2** (Ratio gradient). *Since $\pi_{\text{ref}}$ is fixed, the ratio gradient satisfies*

$$\nabla_\theta r_{i,t}(\theta) = r_{i,t}(\theta)\, \nabla_\theta \log \pi_\theta(y_{i,t} \mid x_i, y_{i,<t}). \tag{25}$$

*Proof.* Let $r = \pi_\theta(y)/\pi_{\text{ref}}(y)$. Differentiating w.r.t. $\theta$ and using $\nabla_\theta \pi_\theta = \pi_\theta \nabla_\theta \log \pi_\theta$ yields the claim. $\qquad\square$

## C.4. Closed-form Token Gradient

Combining Lemma C.1 and Lemma C.2, and treating $A$ and $\epsilon$ as detached during optimization, we obtain the following result.

**Proposition C.3** (Clipped policy gradient). *The gradient of the clipped surrogate satisfies*

$$\nabla_\theta\, s\big(r_{i,t}(\theta)\big) = F_{i,t}^{\text{clip}}(\theta)\, A_{i,t}^{\text{hyb}}\, r_{i,t}(\theta)\, \nabla_\theta \log \pi_\theta(y_{i,t} \mid x_i, y_{i,<t}), \tag{26}$$

*where the clipping gate $F_{i,t}^{\text{clip}}(\theta)$ is given by*

$$F_{i,t}^{\text{clip}}(\theta) = \begin{cases} 0, & A_{i,t}^{\text{hyb}} > 0,\ r_{i,t}(\theta) > 1 + \epsilon(o_i), \\ 0, & A_{i,t}^{\text{hyb}} < 0,\ r_{i,t}(\theta) < 1 - \epsilon(o_i), \\ 1, & \text{otherwise.} \end{cases} \tag{27}$$

*Proof.* In the saturated regions, $s(r)$ is constant and hence has zero gradient. In the unsaturated region, $s(r) = rA$, and the claim follows from Lemma C.2. $\qquad\square$

## C.5. Gradient of the KL Regularizer

Let $h_{i,t} = (x_i, y_{i,<t})$. The KL divergence can be written as

$$D_{i,t}^{\text{KL}}(\theta) = \sum_v \pi_\theta(v \mid h_{i,t})\, \log \frac{\pi_\theta(v \mid h_{i,t})}{\pi_{\text{ref}}(v \mid h_{i,t})}. \tag{28}$$

**Lemma C.4** (KL gradient). *The KL gradient admits the following expectation form:*

$$\nabla_\theta D_{i,t}^{\mathrm{KL}}(\theta) = \mathbb{E}_{v \sim \pi_\theta(\cdot | h_{i,t})} \left[ \left( \log \frac{\pi_\theta(v \mid h_{i,t})}{\pi_{\mathrm{ref}}(v \mid h_{i,t})} + 1 \right) \nabla_\theta \log \pi_\theta(v \mid h_{i,t}) \right]. \tag{29}$$

*Proof.* Differentiate the explicit summation form of $D_{i,t}^{\mathrm{KL}}(\theta)$ and substitute $\nabla_\theta \pi_\theta = \pi_\theta \nabla_\theta \log \pi_\theta$. $\qquad\square$

Consequently, the per-token gradient of the ECHO objective is

$$\nabla_\theta J_{i,t}(\theta) = F_{i,t}^{\mathrm{clip}}(\theta) \, A_{i,t}^{\mathrm{hyb}} \, r_{i,t}(\theta) \, \nabla_\theta \log \pi_\theta(y_{i,t} \mid h_{i,t}) - \beta_{\mathrm{KL}} \, \nabla_\theta D_{i,t}^{\mathrm{KL}}(\theta). \tag{30}$$

## C.6. Implications for Mitigating the Two Challenges via Gradient Allocation

Define the token-wise gradient magnitude (ignoring the KL term) as

$$\mathcal{G}_{i,t} \triangleq \left\| \nabla_\theta \, s\big(r_{i,t}(\theta)\big) \right\| = F_{i,t}^{\mathrm{clip}}(\theta) \cdot \left| A_{i,t}^{\mathrm{hyb}} \right| \cdot r_{i,t}(\theta) \cdot \left\| \nabla_\theta \log \pi_\theta(y_{i,t} \mid h_{i,t}) \right\|. \tag{31}$$

**(i) Suppressing early self-reinforcing overfitting.** By definition of $F_{i,t}^{\mathrm{clip}}(\theta)$, when $r_{i,t}(\theta)$ lies outside the interval $[1 - \epsilon(o_i), \, 1 + \epsilon(o_i)]$ in the direction consistent with the sign of $A_{i,t}^{\mathrm{hyb}}$, the corresponding gradient contribution vanishes due to saturation. Since $\epsilon(o_i)$ is adaptively determined by the trajectory-level tail confidence, overconfident trajectories are assigned a smaller clipping radius, which increases the probability of entering the saturated regime and hence reduces their effective contribution to the update. Therefore, for trajectories that attain spuriously high rewards under noisy pseudo labels and exhibit excessive confidence, the expected gradient magnitude $\mathbb{E}[\mathcal{G}_{i,t}]$ is more strongly upper-bounded, decreasing their likelihood of dominating optimization. Moreover, the KL pull-back term further constrains policy drift, thereby stabilizing training dynamics and preventing premature entropy collapse.

**(ii) Mitigating rollout collapse.** ECHO employs a hybrid advantage of the form

$$A_{i,t}^{\mathrm{hyb}} = A_{g,i}^{\mathrm{grp}} \left( 1 + a S_{i,t} \right), \tag{32}$$

with

$$S_{i,t} = \alpha \, H_{i,t} + \beta \left( 1 - C_{i,t} \right), \tag{33}$$

where $A_{g,i}^{\mathrm{grp}}$ is the within-group trajectory advantage, $H_{i,t}$ is the token entropy at step $t$ on trajectory $o_i$, and $C_{i,t} \in (0, 1]$ is the token-level top-$k$ confidence computed from the logits. This construction induces a token-dependent reallocation of gradient magnitude through the multiplicative factor $1 + a S_{i,t}$. In particular, tokens with high local uncertainty (large $H_{i,t}$) and low confidence (small $C_{i,t}$) yield larger $S_{i,t}$, and thus receive increased gradient weight, encouraging the policy to allocate more learning capacity to decision-critical ambiguous steps rather than overfitting to nearby high-reward neighbors. Conversely, when a trajectory enters a regime where confidence increases and entropy decreases, $S_{i,t}$ becomes smaller and the corresponding gradient amplification diminishes, preventing the update from being dominated by already-certain tokens. As a result, ECHO shifts gradient mass away from persistently uninformative regions and toward uncertainty-resolving positions, and together with entropy–confidence gating and online pruning on the rollout side, reduces the propensity for rollout collapse.

---

**Algorithm 1** ECHO: Entropy–Confidence Hybrid Optimization for Test-Time RL

---

**Require:** Prompt $x$; policy $\pi_\theta$; reference policy $\pi_{\text{ref}}$; rollout size $G$; max length $L$.
**Require:** Windows $W_G, W_T, W_H$; branching params $\{B_{\min}, B_{\max}, \alpha_B, \beta_B, H_{\text{low}}, H_{\text{high}}, s_{\text{branch}}\}$.
**Require:** Pruning params $\{\tau_{\text{prune}}, S_{\text{tail}}, \tau_{\text{tail}}, \delta_{\text{upper}}, S_\Delta\}$.
**Require:** Update params $\{\epsilon_{\min}, \epsilon_{\max}, \kappa, s_{\text{clip}}, \beta_{\text{KL}}\}$; shaping params $\{\alpha, \beta, a\}$.
**Ensure:** Updated parameters $\theta$.
 1: **Warm-up:** estimate $H_{\text{low}}, H_{\text{high}}$.
 2: **Tree rollout with online pruning:** start from root prefix; maintain an active branch set $\mathcal{B}$.
 3: **for** $t = 1$  TO  $L$ **do**
 4:     For each active branch, compute entropy $H_{i,t}$ and token confidence $C_{i,t}$ (Eq. 2).
 5:     Update smoothed confidence $C_t^G$ and $C_{i,t}^{\text{tail}}$ (Eqs. 3–4); compute entropy increment $\Delta H_t$ (Eq. 5).
 6:     Compute branch width $B_t$ by Eq. (6); if $B_t > 1$, fork branches using top-$B_t$ tokens.
 7:     Expand each active branch by one token under $\pi_\theta(\cdot \mid x, y_{i,<t})$.
 8:     Prune branches using low-confidence, tail-decline, and entropy-spike rules.
 9:     **if** all active branches reach a complete answer **then**
10:         **break**
11:     **end if**
12: **end for**
13: Collect $G$ completed trajectories $\{o_i\}_{i=1}^G$.
14: **Majority-vote reward:** compute $\hat{y}$ and set $R_i = \mathbb{I}(\text{answer}(o_i) = \hat{y})$.
15: **Confidence-adaptive clipping:** compute $C_{\text{tail}}(o_i)$ (Eq. 13) and $\epsilon(o_i)$ by Eq. (14).
16: **Hybrid advantages:** compute $A_{g,i}^{\text{grp}}$ by Eq. (15) and $A_{i,t}^{\text{hyb}}$ by Eq. (16)–(17).
17: **Policy update:** update $\theta$ by maximizing $\mathcal{L}_{\text{ECHO}}(\theta)$ in Eq. (18).
18: **return** $\theta$

---

