# OpenReview forum: "ECHO: Entropy-Confidence Hybrid Optimization for Test-Time Reinforcement Learning"
_ICML.cc/2026/Conference — ICML 2026 regular_

### Official Review · Reviewer_P8T1 · 2026-02-14

**Soundness:** 3
**Presentation:** 3
**Significance:** 4
**Originality:** 4
**Overall Recommendation:** 4
**Confidence:** 3

**Summary:**

This paper introduces Entropy-Confidence Hybrid Optimization (ECHO), a framework designed to improve Test-Time Reinforcement Learning (TTRL) for reasoning tasks. Existing TTRL methods typically generate pseudo-labels via majority voting on repeated rollouts. Recent tree-based approaches, such as ETMR, attempt to improve efficiency by branching at high-entropy nodes but suffer from "rollout collapse" (budget concentration on uncertain, ineffective paths) and "self-reinforcing overfitting" (premature convergence to noisy early pseudo-labels).To address this, ECHO proposes a three-fold solution; Entropy-Confidence Hybrid Tree Search, Confidence-Adaptive Clipping, Hybrid Advantage Shaping. Empirical results demonstrate that ECHO outperforms baselines on mathematical benchmarks and multimodal tasks.

**Compliance With Llm Reviewing Policy:**

Affirmed.

**Final Justification:**

The authors have made a genuine effort to address all of my concerns, and I appreciate the additional experiments and clarifications provided. However, after careful consideration, I have decided to maintain my current score.

**Key Questions For Authors:**

## **Questions**

**Q1. Hyperparameter Robustness**: Were the hyperparameters listed in Table 6 kept constant across all experiments (AIME vs. MathVista), or was per-task tuning required to achieve the reported SOTA results?

**Q2. Efficiency vs. Performance**: Could you provide a "performance-per-compute-second" comparison against standard TTRL? Specifically, does the complexity of the tree-management logic negate the gains in sample efficiency in terms of actual wall-clock time?

**Q3. Alternative Reward Signals**: Given the reliance on majority voting, have you considered how ECHO might perform with an external verifier (Outcome Reward Model) instead of a pseudo-label?

**Q4. Comparison to related work**: Recent works like UAA (Uncertainty-Aware Tree Search) [1]  also utilize entropy/uncertainty to guide search budgets. Could you provide a difference of method and a direct performance or efficiency comparison with UAA-style search? Why is ECHO’s "pruning-heavy" approach superior to UAA's "budget-reallocation" approach in the TTRL context?


[1] https://openreview.net/pdf?id=RrLQbXCflj

**Limitations:**

yes

**Strengths And Weaknesses:**

## **Strengths**

**S1. Entropy-Confidence Decoupling**: The core contribution—decoupling entropy (uncertainty) from confidence (path quality)—is a significant conceptual advancement over existing tree-search methods like ETMR. The insight that high entropy alone is a poor signal for branching unless moderated by confidence is well-articulated and effectively addresses the "rollout collapse" problem.

**S2. Robust Policy Update via Confidence-Adaptive Clipping**: The "Confidence-Adaptive Clipping" is a clever reversal of standard RL intuition. By tightening the clipping range for high-confidence trajectories, the authors provide a robust safeguard against the "self-correction" trap where a model overfits to a wrong majority-voted answer.

**S3. Comprehensive Evaluation**: The inclusion of multimodal benchmarks (MathVista/MathVision) alongside traditional text-based math reasoning is commendable. It demonstrates that the ECHO framework is not just a prompt-engineering trick but a fundamental improvement to the TTRL paradigm.

**S4. Soundness**: The ablation studies (Table 3) clearly isolate the gains from the search, clipping, and shaping components, proving that the proposed modules work synergistically.

## **Weaknesses**

**W1. Hyperparameter Complexity**: The method introduces a significant number of hyperparameters (e.g., $W_G, W_T, W_H, \tau_{prune}, \alpha_B, \beta_B$). While the authors provide sensitivity analyses, the sheer volume of "moving parts" raises concerns regarding the ease of deployment and reproducibility in different domains without extensive re-tuning.

**W2. Baseline Capability Threshold**: As noted in the failure analysis (Appendix B.1), the method’s efficacy is tied to the base model's initial reasoning capability. For smaller or weaker models, the "wisdom of the crowd" fails, and ECHO's mechanisms may not overcome poor priors. A clearer discussion on the "minimum viable performance" for ECHO to be effective would benefit the paper.

**W3. Computational Overhead**: While the paper claims sample efficiency, the wall-clock time overhead of maintaining sliding windows and calculating online statistics for every token is not explicitly compared against simpler Best-of-N sampling in the main text.

---

> ### Author Rebuttal · Authors · 2026-03-28
>
> **We thank the reviewer for the suggestion, and our response is as follows:**
>
> ---
> **W1 & Q1**. We make two clarifications. **(1)** ECHO does introduce more hyperparameters than TTRL, but they are not unstructured *moving parts*. They have clear modular roles: **warm-up** for entropy calibration, **branching** for when and how widely to branch, **pruning** for early termination of low-quality trajectories, and **update** for clipping and hybrid advantage shaping. Thus, they serve explicit functions rather than black-box tuning.
>
> **(2)** The main experiments use the unified configuration in **Table 6**, not large-scale per-benchmark tuning. **Tables 1, 2, and 4** show that this shared setup consistently improves performance across text and multimodal tasks, different backbones, and different training/test settings, indicating strong cross-environment robustness without benchmark-specific tuning.
>
> We also acknowledge that not all parameters are equally robust. **Appendix B.1** shows that only a few are closely tied to the main failure modes, especially $\tau_{prune}$ and $W_T$. Therefore, adapting ECHO to a new domain usually requires tuning only a few key parameters rather than the full set.
>
> ---
> **W2**.
> To further discuss the model capability threshold, we conduct supplementary experiments on AIME2024 using three models, and report  Acc together with Label Acc, where Label Acc denotes the accuracy of the pseudo-labels produced by majority voting. The results show a clear pattern: when the backbone is too weak, both TTRL and ECHO fail. For example, for the 0.6B model,  Acc is only 3.3%, Label Acc is just 3.0% and 4.7%, and both methods achieve 0 final performance. This suggests that majority-voted pseudo-labels are too low-quality for ECHO to correct an initially wrong prior.
>
> |Method| ACC (%)|Label Acc (%)|
> |:-:|:-:|:-:|
> |Qwen3-0.6B-Base|3.3|-|
> |+TTRL|0|3|
> |**+ECHO**|0|**4.7**|
> |Qwen3-1.7B-Base|10.0|-|
> |+TTRL|11.5|20.8|
> |**+ECHO**|**13.3**|**28.8**|
> |Qwen3-4B-Base|13.3|-|
> |+TTRL|16.7|30.5|
> |**+ECHO**|**20.0**|**40.9**|
>
> As backbone capability improves, ECHO becomes effective: for Qwen3-1.7B, Acc reaches 10.0% and Label Acc rises to 20.8% / 28.8%; for the stronger Qwen3-4B, the trend is even clearer. Overall, the results suggest a clear capability threshold: ECHO works reliably only when majority-voted pseudo-labels remain positively correlated with true correctness.
>
> ---
> **W3 & Q2**.
> We further compare TTRL, ETMR, and ECHO on Qwen2.5-7B-Base with AIME2024 under the same training configuration, reporting average time per step, average tokens per response, and final performance.
>
> |Method|Avg time/step (s)↓|Avg tokens/example↓|Performance↑|
> |:-:|:-:|:-:|:-:|
> |TTRL|160.7|2782|23.3|
> |ETMR|196.5|1450|24.6|
> |ECHO|210.6|987|30.0|
>
> These results show that ECHO is not the cheapest in raw step time, increasing it by 31.1% / 7.2% over TTRL/ETMR due to extra tree management. However, it reduces average token budget by 64.5% / 31.9% and improves performance by 28.8% / 22.0%, indicating better compute efficiency overall.
>
> ---
> **Q3**.  We have considered replacing majority voting with an external verifier. ECHO is methodologically compatible with this setting: rollout-side scheduling/pruning does not depend on reward source, and update-side clipping or advantage shaping can directly use verifier-provided rewards.
>
> However, replacing majority voting with a verifier still faces two **key challenges**. First, ORM may introduce verifier bias, shifting the problem from pseudo-label bias to verifier bias. Second, introducing an ORM would substantially increase test-time computational cost.
>
> Therefore, if one can design a verifier that is both accurate and efficient, ECHO could in principle benefit further and better mitigate the mislabeling bias introduced by majority voting. This is also a promising direction for our future work.
>
> ---
> **Q4**.  UAA and ECHO differ in both setting and purpose. UAA targets training-free test-time tree search acceleration, using step-level uncertainty to improve search efficiency by reducing redundant exploration. ECHO targets test-time RL, where rollouts also affect pseudo-label generation and policy updates. Accordingly, ECHO uses token-level entropy/confidence and temporal statistics to address high-entropy rollout collapse and early pseudo-label bias in TTRL. Since UAA has not released reproducible code or checkpoints, we have not conducted a direct empirical comparison.
>
> Their pruning roles are also different. UAA’s pruning mainly performs search-side budget compression, retaining a few high-quality candidates to reduce expansion cost. By contrast, ECHO’s pruning in TTRL serves both to avoid wasting rollout budget on high-entropy, low-value branches and to block these branches from contaminating pseudo-labels and later updates. **Thus, in TTRL, ECHO’s pruning is not only for compute efficiency, but also for supervision denoising and update stabilization.**

---

> > ### Author Rebuttal · Reviewer_P8T1 · 2026-04-02
> >
> > Thank you for the detailed and well-organized rebuttal. The authors have made a genuine effort to address all of my concerns, and I appreciate the additional experiments and clarifications provided. However, after careful consideration, I have decided to maintain my current score.

---

> > > ### Author Response · Authors · 2026-04-05
> > >
> > > We **sincerely appreciate** the reviewer's dedication and valuable comments. We are very pleased to hear that our responses have satisfactorily addressed all the issues raised. Thanks to your guidance, the manuscript has been significantly strengthened.

---

### Official Review · Reviewer_vsxM · 2026-03-15

**Soundness:** 3
**Presentation:** 3
**Significance:** 2
**Originality:** 2
**Overall Recommendation:** 4
**Confidence:** 4

**Summary:**

This paper proposes ECHO, a label-free test-time reinforcement learning method that aims to improve self-bootstrapped reasoning by addressing two failure modes in prior TTRL-style methods: rollout collapse under entropy-only tree branching, and self-reinforcing overfitting caused by noisy early pseudo-labels. The method combines three components: an entropy-confidence hybrid tree rollout policy with adaptive branching and online pruning, confidence-adaptive clipping during GRPO-style policy updates, and entropy-confidence hybrid advantage shaping to place more learning weight on uncertain but informative tokens. The paper evaluates ECHO on both text-only reasoning benchmarks and multimodal reasoning benchmarks, showing consistent gains over various baselines across Qwen2.5/3 backbones. The ablations suggest all three components matter, with the tree-search modification appearing especially important, and the analysis section argues that ECHO reduces branch-budget concentration and slows premature entropy collapse during training.

**Compliance With Llm Reviewing Policy:**

Affirmed.

**Final Justification:**

My main concerns have been addressed. I am therefore now leaning positive.

**Key Questions For Authors:**

1. ECHO relies on majority voting for pseudo-labels. In scenarios where the base model is significantly under-capable (e.g., initial accuracy below 20%), the majority may consistently agree on an incorrect reasoning path. Does the confidence-adaptive clipping or hybrid advantage shaping provide a mechanism to recover from such systematic bias?
2. Regarding the online pruning rules, did you observe any cases of false-positive pruning where a path that eventually led to a correct answer was terminated early?

**Limitations:**

No. The paper does not adequately discuss limitations and potential negative societal impact. It briefly states that there are no societal consequences that must be highlighted, which is quite cursory.

**Strengths And Weaknesses:**

Strengths

1. The paper is well-motivated by two key weaknesses in prior test-time RL systems: high-entropy rollout collapse and early pseudo-label bias. It proposes targeted interventions at both the search and update stages to address these issues.
2. The evaluation spans multiple text benchmarks, multiple multimodal benchmarks, multiple Qwen backbones, cross-dataset transfer settings, and IID settings, showing consistent improvements over the baselines.

Weaknesses

1. The paper does not fully disentangle where the gains come from, especially relative to compute and search-budget effects. Some of the method design also feels heuristic, with limited justification for why these particular formulations should be preferred. For example, branch width depends on a hand-designed affine combination of normalized entropy and grouped confidence.
2. Furthermore, the method introduces numerous interdependent parameters and sliding window sizes ($W_G, W_T, W_H, \alpha, \beta$, etc.) and requires a specialized warm-up phase to calibrate thresholds, which may limit its applicability to new domains or model architectures.
3. Implementing the real-time tracking of windowed statistics (moving average entropy and confidence) across multiple parallel tree branches likely adds non-negligible engineering complexity and latency compared to standard chain-of-thought sampling.

---

> ### Author Rebuttal · Authors · 2026-03-28
>
> **Thank you for the reviewer's comments. Below is our response:**
>
> ---
> **W1**.
> ECHO improves performance through better budget allocation and more stable updates. Purely entropy-driven tree search is prone to high-entropy rollout collapse, where a fixed branching budget repeatedly concentrates on a few consecutive high-entropy trajectories, reducing effective branches and pseudo-label diversity. Figure 2 verifies this: compared with ETMR, ECHO reduces the Top-3 budget share from 57.9% to 30.4% and the Top-5 share from 74.5% to 44.2%, reallocating budget from unstable paths to more informative branches.
>
> In addition, Table 3 shows that removing any ECHO component degrades performance, with the largest drop from removing EC-Tree, indicating that high-quality candidate trajectories are critical to the gain. Figures 4 and 5 further show that ECHO mitigates rollout collapse, slows early entropy collapse, and reduces noisy pseudo-label amplification.
>
>
> The **branch-width** is motivated by the complementarity of entropy and confidence. It expands high-entropy, low-confidence regions to encourage exploration, while suppressing high-entropy, high-confidence regions to avoid high-entropy traps. This affine design is monotonic, interpretable, and budget-aware. Its effectiveness is supported by the strong ablation contribution of EC-Tree and by Figures 2 and 5.
>
>
> ---
> **W2**. The warm-up stage does not update model parameters; it only collects entropy statistics to calibrate the entropy scale, avoiding over- or under-branching across models and tasks. Tables 1 and 2 show consistent gains across different model scales, training sources, and both text and multimodal tasks. Table 10 further shows gains under strict IID, indicating that ECHO does not rely on a particular type of shift. We agree that task-specific tuning may further improve performance, but the main results do not rely on test-set-specific tuning and already show strong cross-environment robustness under shared hyperparameters
>
>
> ---
> **W3**.
> Compared with standard CoT sampling, ECHO uses more complex rollout control, but the added cost mainly comes from lightweight statistics and thresholding, not extra LLM forward passes or an additional reward model. Online pruning trades small control overhead for more efficient budget use.  As shown in the table below, On Qwen2.5-7B with training data IME2024, ECHO increases training time per step relative to TTRL and ETMR (+31.1% / +7.2%), but substantially reduces token consumption (−64.5% / −31.9%) and achieves the best performance (+28.8% / +22.0%). This suggests that the added complexity yields **more balanced branch allocation, fewer high-entropy traps and lower token cost, as well as more stable update dynamics**.
>
> |Method|Avg time/step (s)↓|Avg tokens/example↓|Performance↑|
> |:-:|:-:|:-:|:-:|
> |TTRL|160.7|2782|23.3|
> |ETMR|196.5|1450|24.6|
> |**ECHO**|210.6|987|**30.0**|
>
> ---
> **Q1**. **ECHO can mitigate the amplification of such systematic bias.**  **(1)** confidence-adaptive clipping tightens the trust region for trajectories with high tail confidence, thereby limiting early-stage spurious trajectories from dominating policy updates; **(2)** hybrid advantage shaping assigns more gradient mass to tokens that are more informative yet still uncertain, thus alleviating overly rapid contraction and overfitting caused by early pseudo-label bias; **(3)** online pruning on the rollout side further reduces the contamination of subsequent voting and updates by low-quality trajectories, thereby improving candidate quality and lowering the chance of erroneous majority consensus.
> That said, when the backbone is initially too weak to generate almost any correct candidates, majority voting may still become locked onto wrong answers for a long period. In such extreme cases, TTRL cannot recover correct supervision through its internal update mechanism alone.
>
> ---
> **Q2**. **We do observe false-positive pruning.** Appendix B.1 discusses this failure mode: if the pruning threshold is too aggressive, or the window is too short and makes confidence estimation overly sensitive, potentially correct trajectories may be terminated early. For example, on AIME2024, increasing $\tau_{prune}$ from 0.4 to 1.2 reduces Pass@16 from 30.0 to 16.7, confirming that false pruning is real.
>
> To mitigate this issue, ECHO uses tail-smoothed confidence, patience-based pruning triggers, and multi-signal decisions combining grouped confidence, tail confidence, and entropy spikes. These mechanisms cannot fully eliminate false pruning. Appendix B.1 further shows that $\tau_{prune}$ and $W_T$ are the two most relevant hyperparameters: overly strong thresholds or overly short windows amplify false pruning and may weaken or even offset the gains, while proper settings can alleviate it.

---

### Official Review · Reviewer_doJx · 2026-03-15

**Soundness:** 3
**Presentation:** 4
**Significance:** 3
**Originality:** 3
**Overall Recommendation:** 4
**Confidence:** 4

**Summary:**

This paper proposes ECHO, a test-time RL approach that overcomes two challenges, rollout collapse triggered by high-entropy branching and noisy/biased early pseudo-labels, through adaptively controlling branch width via leveraging local entropy and group-level confidence, as well as online confidence-based pruning to avoid high-entropy traps and mitigate collapse.

**Compliance With Llm Reviewing Policy:**

Affirmed.

**Final Justification:**

I remain my initial score.

**Key Questions For Authors:**

- How were the trade-off parameters between entropy and confidence chosen? Can you demonstrate that a single set of hyperparameters works robustly across all the tested environments and shift severities, without requiring test-set tuning?
- If the policy becomes "confidently wrong" due to a severe out-of-distribution shift, the confidence gating might actually accelerate policy degradation rather than prevent it. Have you observed instances where the confidence estimator outputs high confidence for incorrect/suboptimal actions under severe distribution shift? How does ECHO recover from or mitigate this confirmation bias?
- What is the computational cost at test-time, e.g., a quantitative comparison of the inference time between the zero-shot baseline, standard entropy minimization, and ECHO?

**Limitations:**

The paper lacks a discussion on the limitations of the proposed method.

**Strengths And Weaknesses:**

Pros:
- The proposed method is simple and easy to understand, with clear motivations and pipelines to address the claimed challenges.
- The analysis of the two challenges is insightful.
- Experiments are comprehensive, with a good performance improvement of 2-4 points compared to the strongest baseline.

Cons:
- The way of using the entropy or confidence or entropy-confidence for LLM reasoning (post-training RL or test-time RL) is very common now. It's getting a bit visually tiring.
- The balance between entropy and confidence may need careful tuning, especially for new, unseen tasks.
- The paper lacks hyperparameter analysis, e.g., the depth or width values of tree search.

---

> ### Author Rebuttal · Authors · 2026-03-28
>
> **Thank you for the reviewer's comments. Below is our response:**
>
> ---
> **W1**.The core contribution is an entropy–confidence joint optimization mechanism that addresses two key challenges in tree-search-based TTRL: (1) mitigating rollout collapse caused by consecutive high-entropy segments via entropy–confidence joint scheduling and online pruning; (2) suppressing self-reinforcing overfitting induced by early noisy pseudo-labels through confidence-adaptive clipping and entropy–confidence hybrid advantage shaping.
> Together, they alleviate both rollout collapse and confirmation bias.
>
> -----
> **W2 & Q1**.  ECHO does not tune raw entropy or confidence separately for each test task. Instead, it uses a short warm-up phase to calibrate the entropy scale under the current model and task distribution, mapping local entropy into a comparable range. Even with one shared hyperparameter set, ECHO remains stable under substantial distribution shifts. Specifically, Table 1 shows that ECHO consistently outperforms strong baselines across test environments. For example, on unseen AIME2025, ECHO improves Qwen2.5-7B from 23.3 to 33.3. Tables 9 and 10 further show that under stronger cross-domain shifts, including ReClor and math/general reasoning benchmarks, ECHO still outperforms standard TTRL on most test sets.
>
> -----
> **W3**. **We analyze the effects of the pruning threshold $\tau_{prune}$ and window size $W_T$ in Appendix B.1.** The table below further reports sensitivity to tree width and depth. All experiments use Qwen2.5-Math-1.5B trained on AIME2024.
> |$B_{max}$,$B_{min}$|AMC|MATH-500|GPQA|
> |:-:|:-:|:-:|:-:|
> |2,0|35.2|75.1|35.8|
> |4,1|39.8| 80.4 |38.4|
> |6,1|38.9|79.8|37.9|
> |8,1|37.5|78.2|36.8|
> |4,2|38.6|79.1|37.5|
> |4,3|36.8|77.3|36.1|
>
> From the table, $B_{max}=4, B_{min}=1$ gives the best performance. An overly small $B_{max}$ limits exploration in uncertain regions, while an overly large $B_{max}$ introduces redundant branches and reduces budget efficiency. Likewise, an overly large $B_{min}$ enforces unnecessary branching in high-confidence regions and hurts performance. **ECHO therefore benefits from keeping branching width in a proper range to balance exploration and efficiency.**
>
> -----
> **Q2**. ECHO mitigates confirmation bias through two mechanisms:
> **(1) confidence-adaptive clipping**, which assigns a smaller trust-region radius $\epsilon(o_i)$ to trajectories with higher tail confidence, thereby limiting the dominance of high-confidence trajectories in policy updates;
> **(2) entropy–confidence hybrid advantage shaping**, which assigns a smaller trust-region radius $\epsilon(o_i)$ to trajectories with higher tail confidence, limiting their dominance in policy updates
>
> ------
> The table below examines whether the confidence estimator assigns high confidence to incorrect or suboptimal behaviors for Qwen2.5-7B on AIME2024. We measure the proportion of high-confidence erroneous trajectories among voted answers, defined as
> $
> \text{High-conf Err} = \frac{N_{\text{high-conf, wrong}}}{N_{\text{total}}} \times 100%
> $
> where $N_{\text{high-conf, wrong}}$ is the number of trajectories with average confidence above 0.8 but incorrect final answers, and $N_{\text{total}}$ is the total number of generated answers. We also report Clip Radius, the update radius assigned to such trajectories.
>
> |Method|High-conf Err(%)↓|Clip Radius↓|Performance↑|
> |:-:|:-:|:-:|:-:|
> |TTRL|12.8|0.35|23.3|
> |ETMR|10.9|0.20|24.6|
> |ECHO|8.6|0.08|30.0|
>
> From the table above, we can see that the model indeed produces high-confidence but incorrect rollout trajectories, but ECHO significantly reduces the proportion of such trajectories and effectively suppresses their dominant influence on policy updates through a smaller clipping radius, ultimately leading to greater performance improvements.
>
> -----
> **Q3** . The table below compares the training efficiency and final performance of TTRL, ETRM, and ECHO using the Qwen2.5-7B  model on AIME2024. and we report the average training time per step, the average number of tokens per response, and the final performance.
>
> |Method|Avg time/step (s)↓|Avg tokens/example↓|Performance↑|
> |:-:|:-:|:-:|:-:|
> |TTRL|160.7|2782|23.3|
> |ETMR|196.5|1450|24.6|
> |ECHO|210.6|987|30.0|
>
> From the table above, we can see that although ECHO increases the average training time per step by about 31.1% and 7.2% compared with TTRL and ETMR, respectively, it reduces the average token consumption per sample by about 64.5% and 31.9%. Ultimately, ECHO achieves the best performance with the lowest token cost, improving performance by about 28.8% and 22.0%, respectively. This shows that ECHO obtains higher-quality search and update signals with a smaller generation budget.

---

> > ### Author Rebuttal · Reviewer_doJx · 2026-04-03
> >
> > My concerns are well addressed, and I will keep my initial score.

---

> > > ### Author Response · Authors · 2026-04-05
> > >
> > > We are **deeply grateful** for the reviewer's time and constructive feedback. It is encouraging to know that our revisions have fully resolved your concerns. Your insightful suggestions have been instrumental in enhancing the overall quality and clarity of our manuscript.

---

### Decision · Program_Chairs · 2026-04-30

**Decision:**

Accept (regular)

**Comment:**

This paper proposes ECHO, an entropy-confidence hybrid optimization framework for test-time reinforcement learning that aims to address rollout collapse and noisy pseudo-label overfitting in prior tree-based TTRL methods.
Most reviewers agreed that the paper tackles an important problem, is technically sound and well motivated, and reports generally encouraging empirical results. The authors also made substantial efforts during the rebuttal phase and addressed most of the reviewers' concerns. During the reviewer-author discussion phase, all reviewers expressed an overall positive attitude toward the paper. Overall, this paper offers a meaningful contribution to the community. Therefore, I recommend accepting this paper.